

# How Alpine seismicity relates to lithospheric strength

Cameron Spooner [1,2], Magdalena Scheck-Wenderoth [1,3], Mauro Cacace [1], Denis Anikiev [1]

[1]GFZ German Research Centre for Geosciences, Potsdam, Germany
[2]Institute of Earth and Environmental Science, Potsdam University, Potsdam, Germany
[3]Department of Geology, Geochemistry of Petroleum and Coal, RWTH Aachen University, Aachen, Germany

*Correspondence to*: Cameron Spooner (spooner@gfz-potsdam.de)

**Abstract.** Despite the amount of research focused on the Alpine orogen, different hypotheses still exist regarding varying seismicity distribution patterns throughout the region. Previous measurement-constrained regional 3D models of lithospheric density distribution and thermal field facilitate the generation of an observation-based

rheological model of the region. Long term lithospheric strength was then calculated and compared to observed seismicity patterns, showing that the highest strengths within the crust (~ 1 GPa) and upper mantle (> 2 GPa), occur at temperatures characteristic for specific phase transitions (crust: 200 – 400 °C; mantle: ~ 600 °C) with almost all seismicity occurring in in these regions. Correlation in the northern and southern forelands between crustal and lithospheric strengths and seismicity show different patterns of event distribution, reflecting their

different tectonic settings. Seismicity in the plate boundary setting of the southern foreland corresponds to the integrated lithospheric strength, occurring mainly in the weaker domains surrounding the strong Adriatic indenter. However, in the intraplate setting of the northern foreland seismicity instead corresponds to the crustal strength, mainly occurring in the weaker and warmer crust beneath the URG. Results generated in this study are available for open access use to further discussions on the region.

**1 Introduction**

The present-day seismicity distribution is still poorly understood and presently debated in the Alpine orogen and its northern and southern forelands (Deichmann, 1992; Bonjer, 1997; Cattaneo et al., 1999; Singer et al., 2014; Eva et al., 2015). Therefore, any additional constraints on the controlling factors effecting event localisation are of interest. Local models have been used with some success to assess the stress regime on individual faults thereby

offering important insights into the local dynamics driving the observed localisation of seismicity (e.g. Bonjer, 1997). However, and despite its relevance in the current ongoing scientific debate, the cross-correlation between monitored seismicity and its localisation in space with respect to the long-term stress state of the whole lithosphere is still lacking.

Previous works in the region have attempted a quantification of the long-term mechanical state of the lithosphere

mainly relying on 2D sections across the Alpine chain such as the EGT (Okaya et al., 1996) and Transalp (Willingshofer and Cloetingh, 2003) profiles. Relatively few models exist that try to compute lithospheric strength variations across the entire orogen (e.g. Tesauro et al., 2011; Marotta and Splendore, 2014). In this contribution we propose a revision of the current understanding of the long-term lithospheric strength of the lithosphere in the Alpine area in light of higher resolution, region specific 3D geophysical models. To this end, we make use of

recently published results derived from a gravity and seismically constrained structural and density model of the region (Spooner et al., 2019) that has been verified by secondary sources (Magrin and Rossi, 2020) along with a wellbore measurement constrained thermal field (Spooner et al., 2020) to arrive at an updated model of the rheological configuration of the study area.

After a detailed summary of the methodology used to calculate lithospheric strengths, we dedicate the second part

of the manuscript to a critical discussion of the correlation between modelled lithospheric strength variations and



a comprehensive dataset of regional seismicity in order to investigate the potential role that lithospheric heterogeneity plays in shaping the observed localisation of seismicity throughout the Alps and their forelands. In depth analysis leads us to partially revise the main conclusions derived from recent numerical work that attributes the seismicity depth distribution across the region solely to the relatively low plate convergence rate, with

negligible influence from tectonic inheritance (Dal Zilio et al., 2018).

We also quantify the role of viscosity contrasts caused by the presence of a laterally heterogeneous lithospheric configuration in limiting the maximum depth of seismicity through dissipation creep mechanisms. As a result, the 3D distribution of effective viscosities calculated within this work, are able to complement the ongoing debate on the relative impact of glacial isostatic adjustment versus tectonic and mantle dynamic processes as causes of the

observed present-day kinematic state of the Alpine region (e.g. Norton and Hampel, 2010; Chéry et al., 2016; Mey et al., 2017; Sternai et al., 2019).

### 1.1 Geological History

Large scale crustal differentiation within the northern foreland of the Alps (the European plate) primarily results from the Carboniferous age Variscan orogeny (Franke, 2000), such as the juxtaposition of the Moldanubian and

Saxothuringian terrains (Babuška and Plomerová, 1992; Freymark at al., 2017) and the assemblage of crystalline basement presently exposed in the Vosges, Black Forest and Bohemian massifs. Heterogeneity within the Alpine orogen, largely stems from the ongoing collision of the Adriatic plate with the European plate from the Cretaceous to present-day (Handy et al., 2010). Traditionally, the Alpine crust is split up according to its plate of origin prior to orogenesis, such as the European derived Helvetic Alps and the Adriatic derived Southern Alps, that both

represent the proximal domains of their respective plate (Schmid et al., 1989; Schmid et al., 2004). At the surface the present-day boundary of the European and Adriatic derived crust within the Alps broadly occurs at the East-West running Periadriatic Lineament. Ongoing deformation is primarily driven by the convergence of the European and Adriatic plates in northeast Italy (Restivo et al., 2016), where the Adriatic plate is considered to act as a rigid (mechanically stiff) indenter, moving northwards with a radial counter-clockwise rotation against the

weaker European plate (Nocquet and Calais, 2004; Vrabec and Fodor, 2006; Serpelloni et al., 2016). The foreland basins related to the orogen, forming as a result of flexure, are the Po Basin and Veneto-Friuli Plain of the southern foreland and the Molasse Basin of the northern foreland. Also in the northern foreland the Upper Rhine Graben formed as part of the European Cenozoic Rift System in the Eocene (Dèzes at el., 2004). The locations of all relevant tectonic features within the region can be found in Fig. 1.

## 2 Method

Lithospheric structural geometries and densities were sourced from an integration of previous geoscientific datasets in the region by Spooner et al. (2019). Topography and bathymetry (Fig. 1) comes from ETOPO1 (Amante and Eakins, 2009), and the seismically derived lithosphere-asthenosphere boundary (referred to as LAB hereon) originates from Geissler et al (2010). Other sub-surface lithospheric depths were constrained from the use of

numerous published deep seismic surveys (e.g. Brückl et al., 2007; Hetényi et al., 2018), existing structural models of smaller subsets of the study area (e.g. Ebbing, 2002; Przybycin et al., 2014; Freymark et al., 2017) and European plate crustal models (Tesauro et al., 2008; Molinari and Morelli, 2011). Densities were constrained using forward 3D gravity modelling in IGMAS+ (Schmidt et al., 2010, Schmidt et al., 2020) and the global satellite gravity model EIGEN-6C4 (Förste et al., 2014), with the lithospheric layers split into domains of different density to account for

lateral heterogeneity.



The resulting structural and density model (Spooner et al., 2019), with dimensions of 660 km x 620 km (Fig. 1) and a horizontal grid resolution of 20 km x 20 km, represents a 3D structural model of the Alps and foreland regions with the highest spatial resolution among available models and conforms to both seismic and gravity-based observations. Five model layers that represent key lithospheric structural and density contrasts were used for the

rheological calculations: (1) unconsolidated sediments (mostly Quaternary strata); (2) consolidated sediments (mostly Mesozoic strata); (3) upper crystalline crust; (4) lower crystalline crust; and (5) lithospheric mantle. Layer thicknesses and the domains of different density that comprise them are shown in Figures 2 and 3c. Slabs and subduction interfaces are not considered within this work as no consistent model of their geometry or polarity is available for the study area at present (Kästle et al., 2019).

The temperature distribution input to the rheological calculations was obtained from a thermal model of the Alps and their forelands (Spooner et al., 2020) generated from the same structural model utilised here (Spooner et al., 2019). The 3D conductive steady state thermal field was computed using the numerical simulator GOLEM, that can calculate coupled thermal-hydraulic-mechanical processes (Cacace and Jacquey, 2017). Therefore, steady state conductive heat transport was assumed as the main mechanism and specific thermal parameters were assigned to

domains of the structural model, to fit a compiled dataset of measured sub-surface temperatures (Przybycin et al., 2015 and references therein; Freymark et al., 2017 and references; Trumpy and Manzella, 2017). Depth maps of the calculated 200 °C, 400 °C, 600 °C, and 800 °C isotherms are plotted in Figure 4.

The yield strength of the lithosphere (maximum differential stress achievable prior to failure [Goetze and Evans, 1979]) was calculated, taking into account the 3D temperature and pressure state of the system as derived from the

structural and thermal models (Spooner et al., 2019; Spooner et al., 2020). Rheological parameters, also used to calculate structural model layer strengths, were assigned based on the compilation of laboratory measurements (Ranalli and Murphy, 1987) for the dominant lithology interpreted for each layer from observed seismic velocities as well as the modelled density and thermal properties. Parameters used can be found in Table 1. Long term lithospheric strength of the Alps and either forelands were then calculated with a vertical resolution of 100 m,

assuming steady state - secondary creep as the dominant mode of viscous deformation and frictional plastic brittle behaviour following Byerlee's law, using the same methodology described in Cacace and Scheck-Wenderoth (2016).

Byerlee's law (equation 1), a temperature-independent function of confining pressure resulting from increasing density and depth (Byerlee, 1978; Ranalli, 1995) was used to calculate the brittle portion of the yield strength

($\Delta\sigma_b$):

$$\Delta\sigma_b = f_f\,\rho_b g z (1 - f_p) \qquad\qquad (1)$$

where $\Delta f_f$ is the static friction coefficient (set to a constant value of 3 to represent lithospheric stress as per Brace and Kohlstedt [1980]), $\rho_b$ is the bulk rock density, g is the gravitational acceleration, z is the depth (below surface) and $f_p$ is the pore factor (the ratio of the pore pressure to the lithostatic pressure, set here to a constant value of

0.36, representing a fluid density of ~1000 kg m−3 and a rock density of ~2750 kg m−3).

Power law creep (equation 2), representing non-Newtonian, temperature activated deformation of rocks at increasing depth (Karato and Wu, 1993; Burov, 2011), was used to calculate ductile strength ($\Delta\sigma_d$):





$$\Delta\sigma_d = \left(\frac{\dot{\varepsilon}}{A}\right)^{\frac{1}{n}} \exp\left(\frac{Q}{nRT}\right)$$

(2)

where $\dot{\varepsilon}$ is the strain rate (set to a constant $10^{-15}\text{s}^{-1}$, consistent with observed GPS measurements from the region [Houlié et al., 2018]), A is the power-law strain rate, n is the power-law exponent, Q is the activation enthalpy, R is the gas constant, T is the temperature. Tests were made to account for the onset of low temperature crystal plasticity in the lithospheric mantle layer (Peierls creep mechanism [Katayama and Karato, 2008]), however this was found to not affect the ductile strength of the plate.

For the visualisation of maximum rock strength under a constant strain rate at every point in the model, yield strength envelopes (referred to as YSE hereon) showing the differential stress envelope (minimum between $\Delta\sigma_b$-$\Delta\sigma_d$) versus depth were calculated (Brace and Kohlstedt, 1980). The modelled strengths of both the crust and the entire lithosphere were then vertically integrated over their entire thicknesses in order to compare lateral changes in strength throughout the region.

From the above stated relationships, it follows that rates of viscous dissipation in our model can be expressed in terms of a non-linear with temperature effective solid viscosity as ($\eta_{eff}$):

$$\eta_{eff} = \frac{2^{\frac{1-n}{n}}}{3^{\frac{1+n}{2n}}} A^{-\frac{1}{n}} \dot{\varepsilon}^{\frac{1}{n}-1} \exp\left(\frac{Q}{nRT}\right)$$

(3)

The calculated lithospheric strengths and viscosities were then compared to the seismic event catalogue of the International Seismological Centre (International Seismological Centre, 2020) for the study area. The catalogue was filtered to remove events where fixed depths were assigned, where depth error estimates were absent or where the depth error was in excess of 20 % of the event's hypocentre depth (allowing errors of <3 km at a depth of 15 km), to both maximise the accuracy and number of useable events. The catalogue was further filtered to remove events smaller than magnitude 2 in an effort to remove the effects of smaller events 'clustering' around seismic stations observed in similar seismic catalogues (e.g. González, 2016), whilst maintaining coverage across the entire study area. The events used ranged from March 1964 to November 2017, providing a dataset of 4,405 seismic events (shown in Figures 5, 7 and 9) that were then used to interrogate the relationships between lithospheric strength and seismicity throughout the region.

**3 Results**

Across the Alps and their forelands, the pattern of variations in integrated strength of the entire lithosphere (shown in Figure 5a) corresponds closely to the pattern of Moho depth (Figure 3b). The weakest lithosphere (13 $\log_{10}$ Pa m) occurs at the deepest portion of the Alpine crustal root, the largest Moho depth in the study area (55 km), West of the Guidicarie Line. Similarly, the eastern Alps are characterised by both a shallower Moho (45 km) and higher strengths (13.2 $\log_{10}$ Pa m). In agreement with this correlation, both forelands exhibit significantly shallower Moho depths and higher lithospheric strengths than within the orogen. The lithosphere of the southern foreland was also found to be stronger than the northern foreland, with the highest strengths in the study area exhibited on the Apennine plate (13.8 log10 Pa m) and the Adriatic indenter (13.9 log10 Pa m). Similar results have been observed



by previous works modelling the lithospheric strength of the eastern Alps along the Transalp profile (Willingshofer
and Cloetingh, 2003) and in the central Alps along the EGT profile (Okaya et al., 1996) and the entire orogen
(Marotta and Splendore, 2014).

Not all zones of high strength observed in the southern foreland correspond to shallow Moho depths. Whilst the
mechanically strong portions of the Apennine plate occur at the location of the shallowest Moho depth (20 km,
below the Ligurian Sea), the strong Adriatic indenter has some of the largest Moho depths encountered in either
foreland (40 km, south of the Veneto-Friuli plane). Instead the strong indenter is observed to correspond to the
deepest portion of the LAB (140 km, shown in Figure 3d) and to a region of significantly colder lithospheric
temperatures in the Adriatic plate (shown in Figure 4).

The integrated crustal strength (shown in Figure 5b) also positively correlates with temperature, being highest
(13.175 $\log_{10}$ Pa m) in the South East of the Adriatic indenter where all isotherms are deepest (Figure 4). North of
the Periadriatic lineament and West of the Guidicarie Line, isotherm depths are consistently shallower than in the
southern and eastern Alps. Consequently, the northern and western Alps feature lower crustal strengths (13 $\log_{10}$
Pa m) than the Alpine crust in the south and east (13.125 $\log_{10}$ Pa m) a finding also observed by Marotta and
Splendore (2014). Where isotherms are raised below the URG, crustal strengths are also the lowest (12.925 $\log_{10}$
Pa m).

The distribution of seismic event epicentres in the southern foreland strongly correlates spatially with the computed
integrated lithospheric strength (Figure 5a) and not with crustal strengths, as most events occur in the weak
lithosphere surrounding the more rigid Adriatic indenter or Ivrea body. Within the northern foreland no correlation
is observable between seismicity localisation and lithospheric strength, with epicentres instead corresponding
closely to the weaker portions of the crust (Figure 5b) around the URG in the west of the Moldanubian domain.
Both integrated lithospheric and crustal strength maps portray lower strengths within the Alps proper, North and
West of the Guidicarie Line, corresponding to the location of the majority of Alpine seismicity.

Cross sections showing variations in differential strength (minimum between brittle strength and ductile strength)
with depth and the relation to seismicity are plotted in Figure 6. In line with previous works (Okaya et al., 1996;
Willingshofer and Cloetingh, 2003; Marotta and Splendore, 2014), all cross sections show that the majority of
seismicity occurs within the strongest region of the upper crust (~ 1 GPa), between 200 °C and 400 °C, with a
largely aseismic and weaker lower crust mechanically decoupling the crust from the lithospheric mantle.
Seismicity deeper than the upper crust is however present in regions where the upper lithospheric mantle is cooler
than 600 °C and strong (> 2 GPa), as shown in cross sections 1 and 3. Regions seen in Section 1 and 2 characterised
by a stronger lower crust (~ 1 GPa) also show seismicity to a greater depth. Section 4, which runs West to East
through the centre of the orogen, does not portray a strong lower crust or upper lithospheric mantle, and shows
Moho temperatures consistently higher than 600 °C, exhibiting no seismicity outside of the upper crust.

The pattern of variations in integrated effective viscosity of the lithospheric mantle (shown in Figure 7) are the
same as the integrated strength distribution of the whole lithosphere (Figure 5a), with higher strengths analogous
to higher viscosities, corresponding closely to the Moho depth (Figure 3b). Seismicity is therefore observed in the
orogen and southern foreland to spatially correlate to regions of lower viscosity such as the Alpine root (20.6 $\log_{10}$
Pa s) that surround the higher viscosity blocks such as the Ivrea Zone, Adriatic Indenter and Apennine plate (22.2
$\log_{10}$ Pa s). In the Northern Foreland the opposite is observed, with seismicity occurring in the region of highest
viscosity surrounding the URG (20.8 $\log_{10}$ Pa s). Similar results, showing that viscosities below the crustal root
must be lower than below the forelands have also been modelled by Chéry et al. (2016), in order to fit the observed





trends of glacial rebound.

Variations of effective viscosity with depth in relation to seismicity are shown in the cross sections on Figure 8. As previously observed, the majority of seismicity occurs in the upper crust, which largely corresponds here to effective viscosities of 23.5 $\log_{10}$ Pa s or higher. Viscosities for the lithospheric mantle tend to be between 19 – 23 $\log_{10}$ Pa s and for the lower crust between 21 – 23 $\log_{10}$ Pa s with both largely aseismic across the region. Regions

where seismicity does occur deeper than the upper crust also correspond to regions of the lower crust or lithospheric mantle where effective viscosities are also in excess of 23.5 $\log_{10}$ Pa s, such as in the South East of Section 1 and the North of Section 3.

### 4 Discussion

#### 4.1 Mechanical Strength

The strongest regions at depth in the study area correlate with the 600 °C isotherms within the upper lithospheric mantle, corresponding to a phase change in mantle rocks (Boettcher et al., 2007, McKenzie et al., 2005). Additionally, the map of the integrated lithospheric strength (Figure 5a) portrays a positive inverse correlation with respect to the Moho depth (Figure 3b), allowing Moho depth throughout the region to be used as a first order estimate for the relative strength of the whole lithosphere. Based on these findings we can conclude that the

lithospheric mantle is both the largest contributor to the overall computed lithospheric strength variations and is also highly influenced by the temperature configuration across the entire orogen foreland system, thereby expanding upon previous findings derived from, but limited to, the central Alps (Okaya et al., 1996).

The integrated crustal strength (Figure 5b) amounts only to a small portion of the total lithospheric strength (Figure 5a) except in locations where the crust is at its thickest such as the crustal root of the orogen (Figure 3a). Under

this area, the pattern of crustal strength distribution equals the whole plate strength distribution. The presence of a weak and thick crust North of the Periadriatic lineament contributes to a significant weakening of the lithosphere underneath this domain, a feature that was also noticed by previous work (Marotta and Splendore, 2014). To further deepen the discussion about the implications derived from the thermo-rheological model on the seismicity distribution within the area we also note that Alpine events mostly occur beneath this domain, North and West of

the Guidicarie fault, consisting mainly of the Helvetic nappes, where the crust is both warmer and weaker. This is part of a broader observed trend of West to East mechanical strengthening within the Alpine crust, that results in significantly less seismicity in the Eastern Alps, explained by a deepening of LAB topography (Figure 3d) and therefore a lower geothermal gradient.

The temperature distribution throughout the region is primarily a function of lithospheric composition, with the

relative contribution of model layers to the heat budget varying in response to their specific thermal properties and relative volume. Features such as a shallow thermal LAB, here derived from a global tomographic model (Schaeffer and Lebedev, 2013), a higher percentage of felsic (radiogenic) upper crust to mafic lower crust or the presence of thick insulating sediment deposits have been previously shown to raise the geothermal gradient within the study area (Spooner et al., 2020). We therefore expect that these specific features would also exert a first-order

control on the resulting mechanical configuration of the lithosphere and thereof to the seismicity distribution.

#### 4.2 Relation to Seismicity

The northern and southern forelands of the Alps display a markedly different pattern of seismicity in terms of their epicentre locations, potentially reflecting the different tectonic settings and driving mechanisms at play within each foreland. In the southern foreland seismicity primarily occurs at the boundaries of the European, Adriatic and

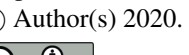



Apenninic plates (e.g. Chiarabba et al., 2005). These locations mark a relatively sharp gradient in modelled lithospheric strength and effective viscosities from 13.9 $\log_{10}$ Pa m and 22.2 $\log_{10}$ Pa s within the plate to 13.2 $\log_{10}$ Pa m and 20.8 $\log_{10}$ Pa s along its edges, respectively. These mechanically stiff and rheologically strong lithospheric blocks move independent of one another (e.g. Nocquet and Calais, 2004). Therefore, it stands to reason that seismicity in such a setting would localise at the boundaries of these rigid lithospheric blocks (Figure 5a and

Figure 7). The situation differs in the northern foreland, where seismicity occurs within an intraplate setting (e.g. Bonjer, 1997) of the European plate and where the upper mantle is not seismogenic. It therefore seems logical to assume that under these tectono-thermal conditions the weaker regions of the crust would accommodate the majority of seismicity as depicted in Figure 5b, where the lithospheric mantle remains relatively stable throughout the northern foreland.

It is nowadays established that temperature is an important variable for determining the depth of the lithospheric seismogenic domain. This was noted in the seminal study of Sibson (1982) that demonstrated a correlation between intraplate seismicity focal depths and surface heat flow distribution, with shallow seismicity in areas of high surface heat flow and vice versa. The existence of an inverse correlation between heat flow and focal depth is easy to explain when considering the homogeneous configuration of oceanic plates. However, it is challenging to

extrapolate these results to continental intraplate regions where the log-linear age-temperature relationship does not apply and the thermal state is a complex function of the history of a heterogeneous plate over geological times. In such cases, a conservative estimate for the lower bound of the seismogenic zone can be derived based on a quasi-static thermodynamic description of the continental plate characterized by a non-Newtonian fluid-like rheology descriptive of its most abundant constitutive minerals. By relying on such a description, a maximum in

the static strength would correspond to a particular value of the system's internal energy, which can then be described by the temperature at its maximum dissipation potential. Under this assumption, one would expect that to a first-order, within the study area the 600 °C isotherm, which represents this transition for olivine rich lower crustal and mantle rocks (Boettcher et al., 2007, McKenzie et al., 2005), would both represent a maximum lower bound to seismicity and also the highest strengths in the lower crust and upper lithospheric mantle.

The majority of observed seismicity occurs between the 200 °C and 400 °C isotherms, representing the strongest portion of the upper crust (up to ~1 GPa). Deeper in the crust, higher temperatures result in a gradual decrease in the plate strength and subsequent aseismic behaviour. These observations can be taken as indicative of the brittle-ductile transition within the crust, that provides a conservative estimate to the lower bound of the seismogenic zone in that area. Willingshofer and Cloetingh (2003) performed an end-member sensitive analysis for the

lithospheric strength along the Transalp section of the eastern Alps in terms of considering either a strong or a weak crust. The main conclusion derived from their study was that a model portraying a strong crust provides a better fit with the seismicity. From their analysis they determine that the brittle-ductile transition occurs between 9 and 14 km. By plotting the depth level above which 95% of seismic events occur (Figure 9), our results, based on more up-to-date 3D structural and thermal model, estimate that the brittle-ductile boundary within the Alps

occurs at ~ 20 km depth, providing a better fit with the depths of recorded seismicity and adding validity to the model setup utilised here.

    Additionally, whilst the majority of seismicity occurs in the upper crust, observations also show that seismicity in deeper layers occurs in both the northern and southern forelands (e.g. Bonjer 1995; Chiarabba et al., 2005). From the differential stress cross sections shown here (Figure 6), it can be discerned that hypocentre depths of deeper

seismicity vary with temperature, occurring where upper lithospheric mantle temperatures are ~ 600 °C or cooler.



This thermal configuration leads to the presence of a relative weak lower crust, mechanically decoupled from and sandwiched between the upper crust and a strong upper mantle (more than 2 GPa), thus providing a first-order explanation to the deepening of seismicity in the area. Seismicity within the lower crust is found to only occur in domains of higher strength (~ 1.4 GPa), though within the majority of the region the lower crust is observed to be

largely weak and therefore aseismic. This is particularly evident from the effective viscosity cross sections (Figure 8) depicting how the lower crust mechanically decouples the upper crust from the lithospheric mantle, with effective viscosity values (22.5 log10 Pa s) indicative of a ductile regime and fitting well with the general lack of observed lower crustal seismicity.

The unimodal pattern of seismicity beneath the Alps being limited to the upper crust contrasts to the bimodal (crust

and upper mantle) seismicity pattern found in other orogens worldwide, such as the Himalaya. Based on this observation, a recent modelling study by Dal Zilio et al. (2018) advanced the hypothesis of a structural correlation between plate convergence rates and seismicity distribution alone. The analysis of Dal Zilio and co-authors is based on a linear correlation between convergence rates and the resulting thermal configuration of the orogen, with faster rates resulting in a colder orogen and therefore in a more widespread seismogenic brittle domain. A

major limitation of this reasoning is that it does not take into account the role of crustal inheritance. Whilst results presented here mostly agree with the unimodal nature of seismicity throughout the crust, they also suggest that the lithospheric makeup of the region such as crustal, and lithospheric thickness have a first-order effect on the location of seismicity in the region via their control on the lithospheric thermal field. In addition, relating the overall distribution of seismicity within the Alpine region to the background tectonic convergence rates cannot reconcile

the diversity in the observed seismic style across the whole orogen. While the eastern Alps are characterized by seismicity showing mainly dilatational faulting, the western and central Alps portray mainly compressional seismicity. Therefore, it would be difficult to relate this difference in style to a common geodynamic process. Indeed, there is evidence from present day uplift rates that the seismicity in the western Alpine domain could be instead related to still ongoing viscous relaxation from the waning of the last ice cap (e.g. Norton and Hampel,

2010; Chéry et al., 2016; Mey et al., 2017; Sternai et al., 2019). In this regard a word of caution comes from the uncertainty in the mantle and (lower) crustal viscosities input in these studies. Past works have adopted values as low as ~20 $\log_{10}$ Pa s for mantle viscosity (Norton and Hampel, 2010; Chéry et al., 2016; Mey et al., 2017) whilst more recent work has made use of higher values ~22 $\log_{10}$ Pa s for the Alpine lithospheric mantle (Sternai et al., 2019). The values derived in the present study, ranging from ~21 $\log_{10}$ Pa s beneath the orogen to ~22 $\log_{10}$ Pa s

in the forelands compare favourably to those estimates. In addition, our model favours the presence of lower viscosities below the orogen domain proper in comparison to below the forelands, a result that agrees with the main conclusions derived from Chéry et al. (2016). This last result confirms how the pattern and style of seismicity within the complex Alpine area cannot be related to a single geodynamic parameter, such as convergence rates, but should be considered as a natural outcome of a rather complex crustal structure developed during the whole

orogenic cycle in an ongoing plate tectonic setting.

### 4.3 Slab Influence

The temperature present at the maximum depth of seismicity (Figure 9) is plotted in Figure 10a, showing that most mechanically strong portions of the plate, whether within the crust or the upper lithospheric mantle, are effectively bounded by the depth of the 600 °C isotherm previously discussed to represent the maximum temperature of

seismicity. This trend also visible in the various cross sections through the region (Figure 6). In thick felsic crustal regions that also lie above a weak lithospheric mantle, such as the crustal root of the orogen, maximum depths of



seismicity are significantly shallower than on the forelands and as such maximum temperatures of seismicity are also significantly lower at ~ 350 °C. We do however note regions where the maximum temperature of seismicity greatly exceeds 600 °C, corresponding to the presence of both actively subducting and previously subducted slabs,

shown as high velocity features at a depth of 100km (Figure 10b) from a recent shear wave velocity model of the region (El-Sharkawy et al., 2020). As the thermal field utilised here (Spooner et al., 2020) is calculated as static and steady state, the cooling effect of subduction zones, which is still largely unquantified, is not incorporated. Therefore, the possibility of seismicity occurring in these regions at higher temperatures than expected is anticipated, as previously discussed in Spooner et al., (2020). This effect is most pronounced at the location of the

active Apennine subduction zone where maximum temperatures of seismicity appear to approach 1000 °C, however regions where seismicity above 600 °C are also noticed below the Alps, corresponding to the location of Alpine slabs, indicating that these frozen in subduction zones may still be having a thermal effect on the lithosphere.

### 5 Summary

In this work we have computed the long-term lithospheric yield strength for the Alpine regions and its forelands by using available up to date structural, density and thermal input data. Variations in the strength of the upper lithospheric mantle exert the largest influence on the strength of the whole lithospheric column, with crustal strengths only contributing significantly to the whole plate integrated strength beneath the orogen proper, where the crust is thickest (55 km). The strengths, whether in the crust or mantle, are largely temperature dependant, with

upper lithospheric mantle temperatures controlled by Moho depth and crustal temperatures by thermal LAB depth, thickness of the radiogenic felsic upper crust and thickness and distribution of insulating sediments and.

The results from the thermo rheological modelling exercise has shed light on the relationship between background seismicity and resolved lithospheric strength variations in the region. We have been able to demonstrate how the occurrence of crustal seismicity in the study area is influenced by several factors acting at different scales with

inherited geological crustal and upper mantle structures exerting a primary control in the seismicity distribution and style. The highest yield strengths within the crust (~ 1 GPa) and upper mantle (> 2 GPa) occur at temperatures interpreted as phase transitions (crust: 200 – 400 °C; mantle: ~ 600 °C) with almost all seismicity occurring in these regions. We also note the presence of a weak lower crust (< 1 GPa) that mechanically decouples the upper crust and lithospheric mantle across the entire region. Therefore, the lower crust appears largely aseismic, likely

due to seismic energy being dissipated by ongoing creep in regions where effective viscosities are lower. Both active present day and frozen-in subducting slabs are also shown to significantly influence the maximum depth of seismicity possible above them, furthering the argument that lithospheric inheritance and heterogeneity within the region, are key components to explain the regional distribution of seismicity.

In the Alps, seismicity correlates spatially with a weaker crust and lithosphere, such as the Helvetic Alps. Such a

clear distinction cannot be derived uniquely for both forelands, each showing a different pattern of seismic distribution, likely reflecting their different tectonic settings. In the southern foreland seismicity preferentially occurs across boundaries between rigid lithospheric blocks, such as the strong Adriatic indenter, whilst in the northern foreland seismicity localises beneath domains of crustal weakness as in the URG.

### Data Availability

Data will shortly be made available through GFZ Data Services.



**Author Contributions**

Mauro Cacace and Denis Anikiev developed tools for calculations in the modelling workflow and plotting of results. Magdalena Scheck-Wenderoth advised on the entire workflow and the interpretation of results. Cameron

Spooner carried out the modelling work and prepared the paper with contributions from all co-authors.

**Competing Interests**

The authors declare that they have no conflict of interest.

**Special Issue Statement**

This manuscript is submitted to the special issue: New insights on the tectonic evolution of the Alps and the

adjacent orogens.

**Acknowledgments**

The authors would like to thank the Deutsche Forschungsgemeinschaft (DFG) for funding the Mountain Building Processes in Four Dimensions (4-D-MB) SPP that this work was produced as part of.

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





**Tables**

| Model layer and representative rheology used | Power-law strain rate $A$ $(Pa^{-n}\, s^{-1})$ | Power-law Exponent $n$ - | activation enthalpy $Q$ $(J\, mol^{-1})$ |
|---|---|---|---|
| Sediments (Granite, wet) | 7.94E-16 | 1.9 | 1.37E+05 |
| Upper Crust (Quartzite) | 2.51E-24 | 2.4 | 1.56E+05 |
| Lower Crust (Diabase) | 8.04E-25 | 3.4 | 2.60E+05 |
| Mantle (Olivine) | 4.00E-12 | 3 | 5.40E+05 |


Table 1 – Representative lithologies and rheological parameters (from Ranalli and Murphy, 1987) used for modelling the layers of the structural model.




**Figures**

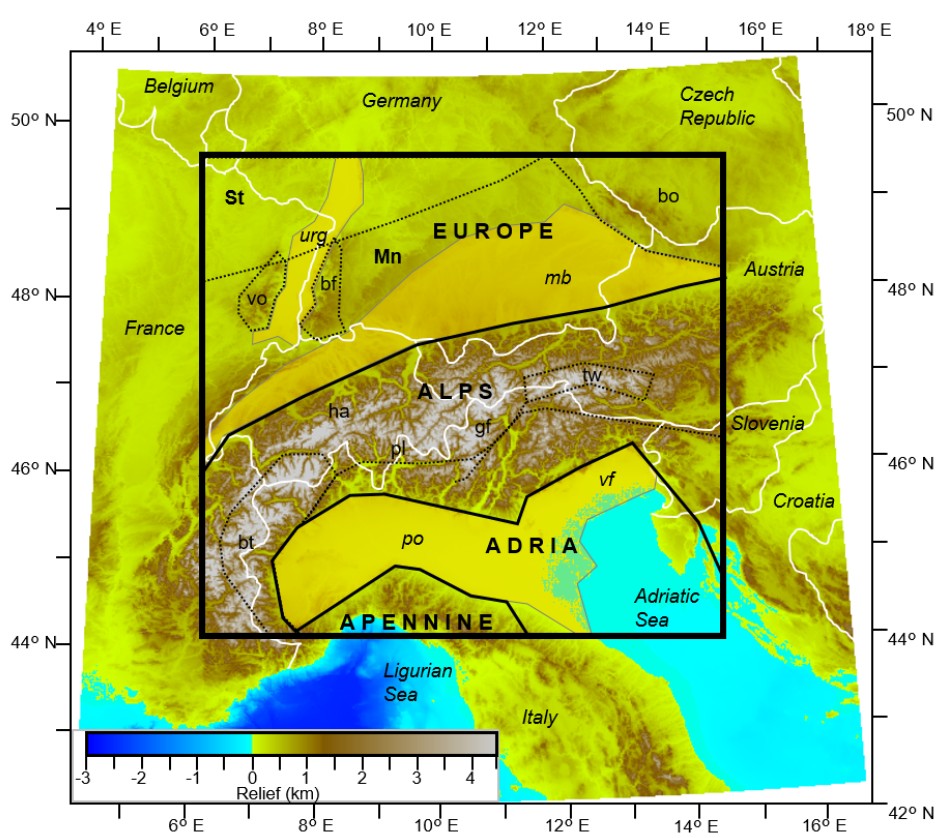

- Figure 1. Topography and bathymetry from ETOPO 1 (Amante and Eakins, 2009) shown across the Alpine region with the key tectonic features overlain. Study area is indicated with a black box. Solid black lines

demark the boundaries of the weakly deformed European and Adriatic plates, the location of the Apennine plate is also marked. Yellow areas bound by a solid grey line indicate the extent of sedimentary basins (urg – Upper Rhine Graben; mb – Molasse Basin; po – Po Basin; vf – Veneto-Friuli Plain). Dotted black lines indicate the extent of other tectonic features within the model (St – Saxothuringian Variscan domain; Mn – Moldanubian Variscan domain; bo – Bohemian Massif; vo – Vosges Massif; bf – Black Forest Massif; ha – Helvetic Alps; tw – Tauern

Window; gf – Giudicarie Fault; pl – Periadriatic Lineament; bt – Brianconnais Terrane). The Adriatic Sea is marked as (AS) in further figures.


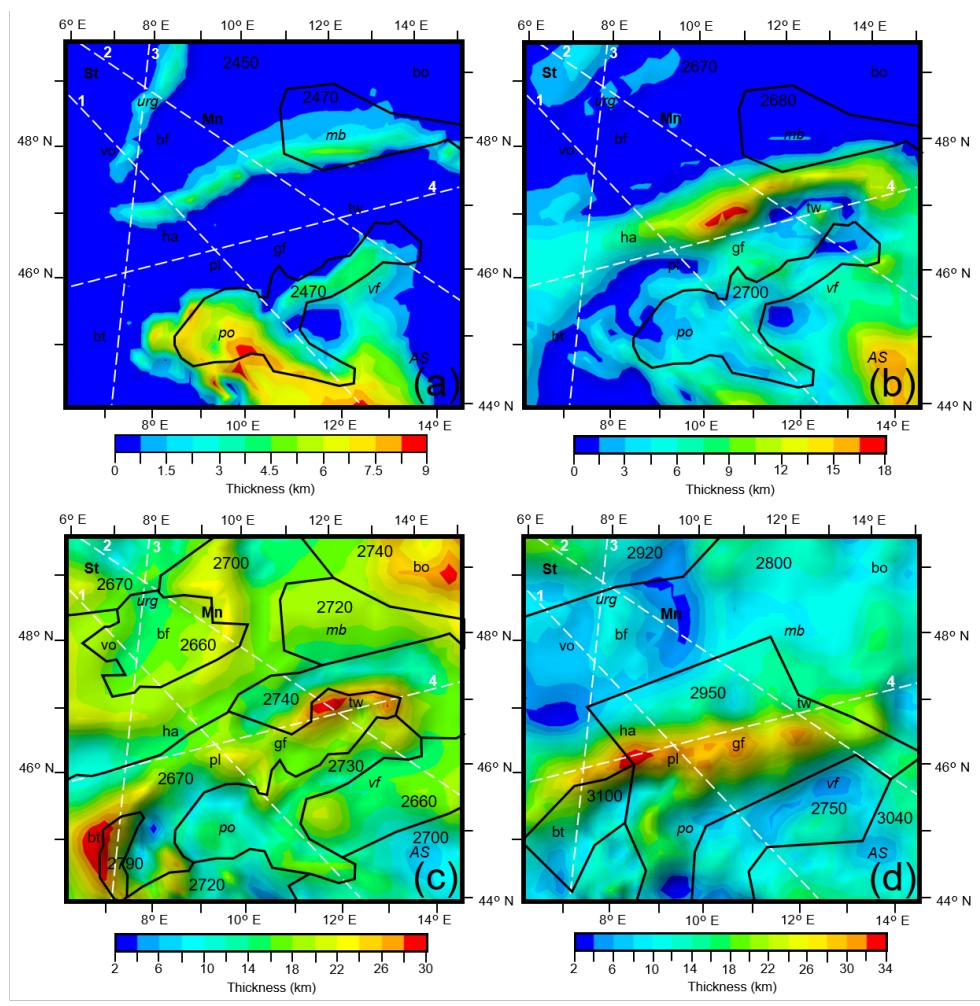

-        Figure 2. Thickness of a) unconsolidated sediments, b) consolidated sediments, c) the upper crust and d)
the lower crust across the modelled area. Locations of key tectonic features are overlain (abbreviations shown in
Fig. 1a caption). Cross sections 1, 2, 3 and 4 from Figures 6 and 8 are marked with white dashed lines.

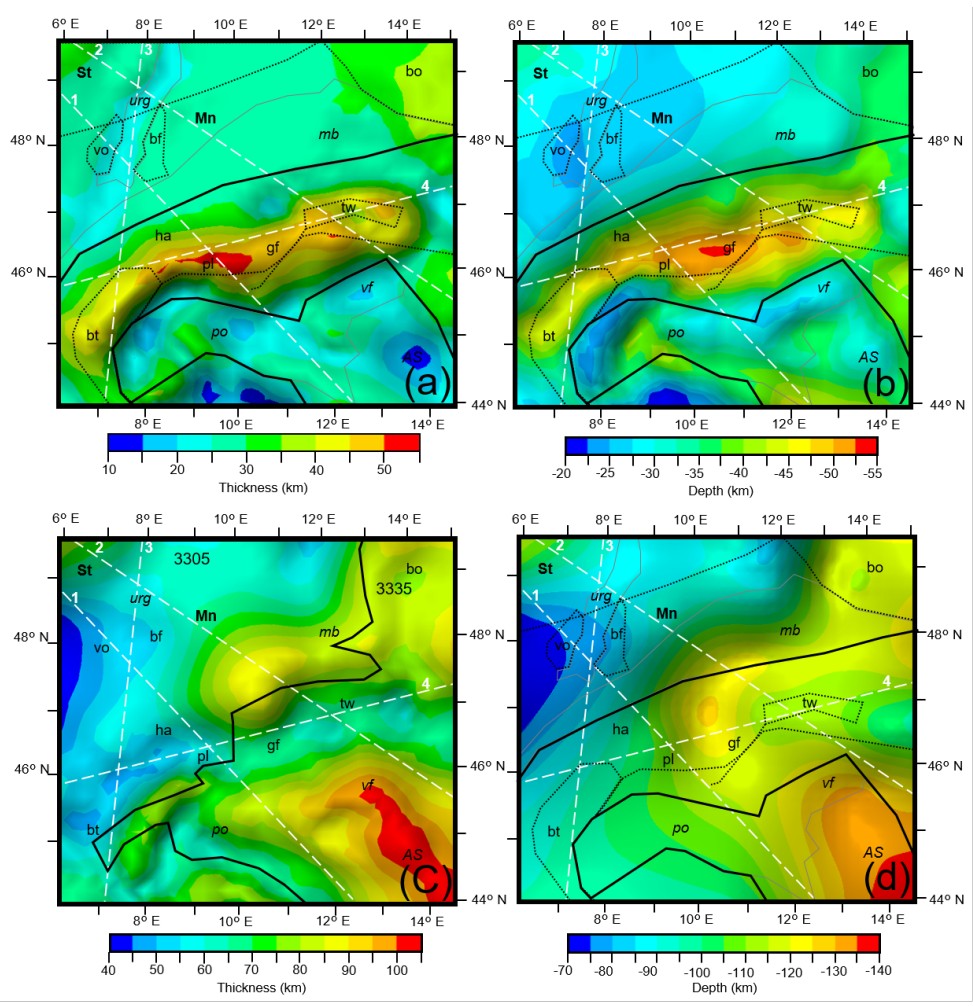

- Figure 3. a) Thickness of the entire crust. b) Moho depth. c) Thickness of the lithospheric mantle. Density domains required during modelling are outlined in black with the density used for each (in kg/m3) shown within. d) LAB depth (from Geissler et al., 2010). Locations of key tectonic features are overlain (abbreviations shown in Fig. 1a caption). Cross sections 1, 2, 3 and 4 from Figures 6 and 8 are marked with white dashed lines.




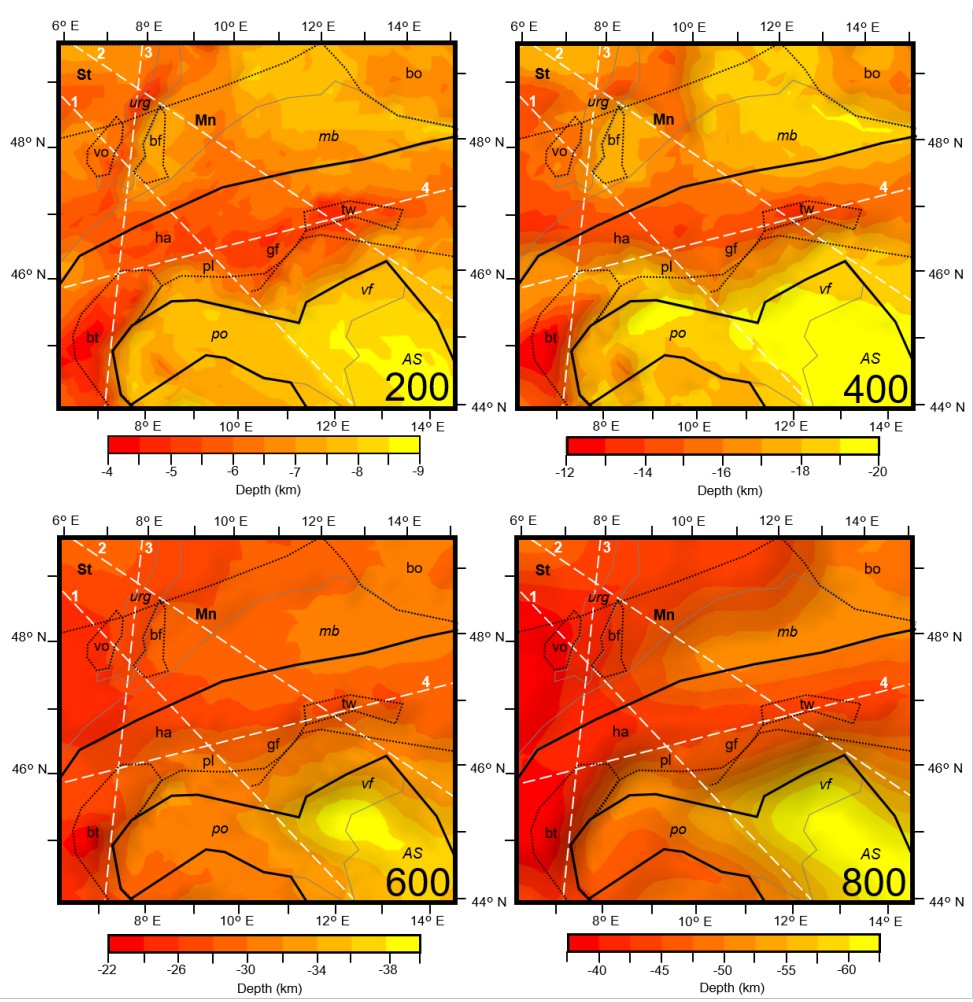

- Figure 4. Depths to the 200 °C, 400 °C, 600 °C and 800 °C isotherms across the modelled area. Locations of key tectonic features are overlain (abbreviations shown in Fig. 1a caption). Cross sections 1, 2, 3 and 4 from Figures 6 and 8 are marked with white dashed lines.






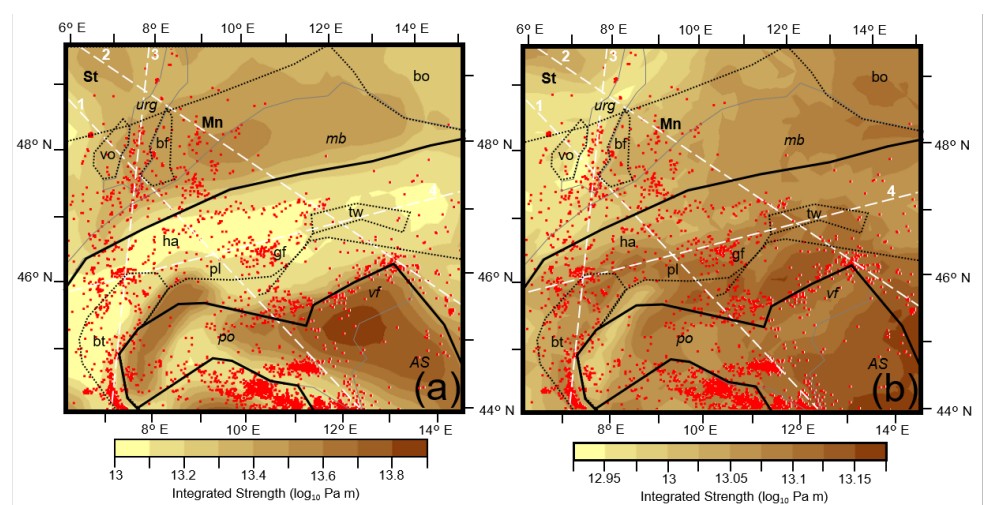

-      Figure 5. Integrated strength of a) the lithosphere and b) the crust across the modelled area with seismicity > M2 shown in red dots. Locations of key tectonic features are overlain (abbreviations shown in Fig. 1a caption). Cross sections 1, 2, 3 and 4 from Figures 6 and 8 are marked with white dashed lines.







- Figure 6. Cross sections 1, 2, 3 and 4 (Locations shown in Figures. 2, 3, 5 and 7) showing the variation of differential stress with depth throughout the lithosphere. The top of the upper crust, lower crust and Moho are shown as solid black lines. Isotherms up to 1000 °C are overlain as dashed blue lines and seismicity > M2 that lay within 20 km distance of the section has been marked as red dots.

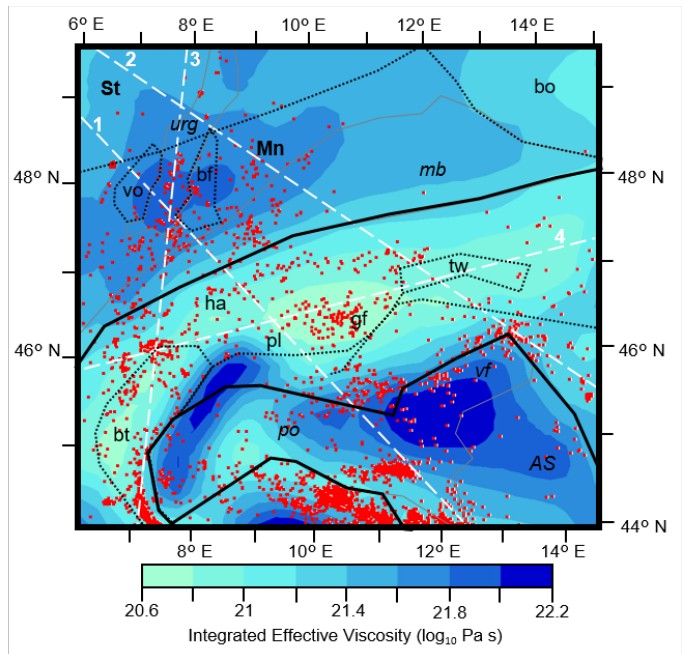

- Figure 7. Integrated effective viscosity of the lithospheric mantle across the modelled area with seismicity > M2 shown in red dots. Locations of key tectonic features are overlain (abbreviations shown in Fig. 1a caption). Cross sections 1, 2, 3 and 4 from Figures 6 and 8 are marked with white dashed lines.






Figure 8. Cross sections 1, 2, 3 and 4 (Locations shown in Figures. 2, 3, 5 and 7) showing the variation
of effective viscosity with depth throughout the lithosphere. The top of the upper crust, lower crust and Moho are
shown as solid black lines. Seismicity > M2 that lay within 20 km distance of the section has been marked as red
dots.

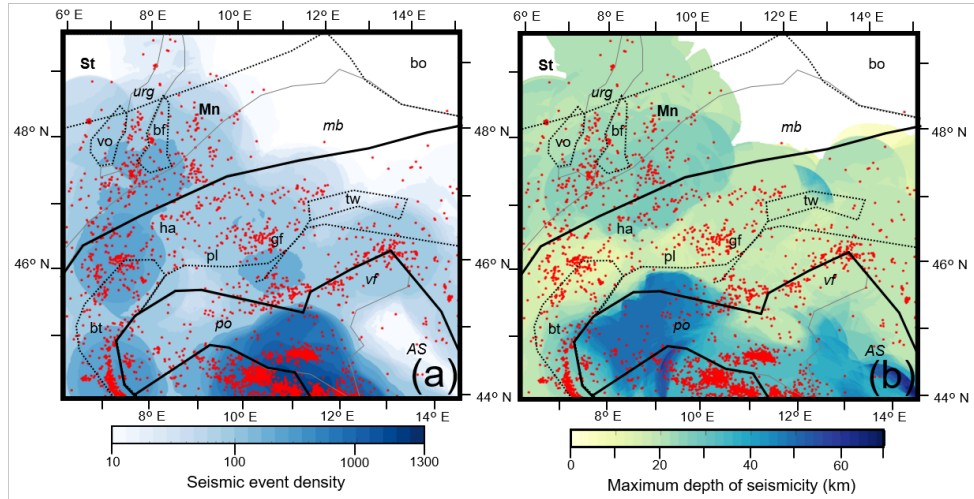

Figure 9. a) Seismic event density and b) Maximum depth of seismicity (depth above which 95% of
events occur) for events > M2 within a radius of 75 km of each grid point. Locations of key tectonic features are
overlain (abbreviations shown in Fig. 1a caption).

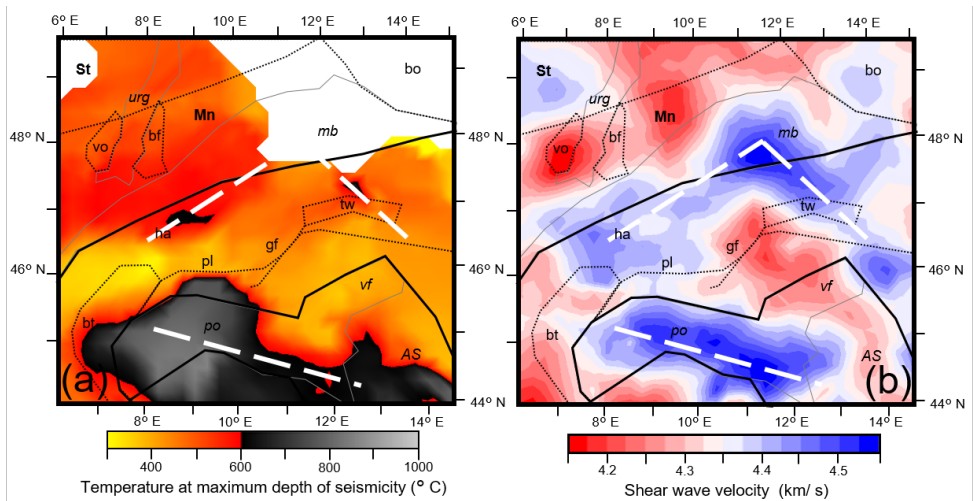

Figure 10. a) Temperatures that correspond to the maximum depth of seismicity (Figure 9b). b) Shear
wave velocity model at 100 km depth from El-Sharkawy et al. (2020). Locations of slabs are highlighted with
white dashed line. Locations of key tectonic features are overlain (abbreviations shown in Fig. 1a caption).