# Peer review of "How Alpine seismicity relates to lithospheric strength"

_Solid Earth, 2020_

## Referee Comment (RC1) · Anonymous Referee #1 · 29 Jan 2021

Dear editor,

Thank you for considering me as a reviewer for the manuscript entitled "How Alpine seismicity relates to lithospheric strength" by Spooner and co-authors. Please, find my comments below.

General comments

In this ms the authors investigate the relation between the spatial distribution of seismicity and the strength of the lithosphere in the Alps. They propose a new 3D model for the yield strength of the lithosphere, based on recent thermal and structural models, which is of great interest for better constraining the geodynamics of this complex area. The comparison of their results with the spatial distribution of earthquakes is also

relevant.

Although well written and illustrated, this ms has to be revised before being published. One concern is the lack of discussion regarding the representativeness of the seismic catalog used in the study. Although the authors have selected the events according to the location error and the magnitude, the limit of the catalog extracted from ISC and the potential bias linked to the seismic station distribution aren't discussed. This should be addressed, in order to be able to compare the modeled strength of the lithosphere with the epicenters and the depth distribution of the events. More generally, it's also essential to add a synthesis of the knowledge of the factors controlling the seismicity localization in the area, based on previous studies (e.g. Schmid and Kissling, 2000; Singer et al., 2014; Thouvenot et al., 2016) to the ms. Actually, it would help to better contextualize the study and highlight its importance.

A large part of the description of the results and the discussion refers to the spatial variability of several parameters (viscosity, integrated strength, maximum depth of seismicity, etc.). These sections are however difficult to follow because this spatial distribution isn't explicit enough with respect to the areas of interest. Using systematically the annotation in the figures (AS, vo, urg,. . ..), or naming clearly the regions would greatly help to understand the reasoning of the authors. E.g. in the sentence L314-316 (" We do however note regions where the maximum temperature of seismicity greatly exceeds 600°C, corresponding to the presenceof both actively subducting and previously subducted slabs, shown as high velocity featuresat a depth of 100km(Figure 10b) from a recent shear wave velocity model of the region (El-Sharkawy et al., 2020)"), it's unclear which regions the authors are referring to.

Schmid, S. M., and Kissling, E. (2000), The arc of the western Alps in the light of geophysical data on deep crustal structure, Tectonics, 19( 1), 62– 85, doi:10.1029/1999TC900057.

Singer, J., Diehl, T., Husen, S., Kissling, E. and Duretz, T. Alpine lithosphere slab

rollback causing lower crustal seismicity in northern foreland. Earth and Planetary Science Letters, 397, pp.42-56, https://doi.org/10.1016/j.epsl.2014.04.002, 2014.

Thouvenot, F., Jenatton, L., Scafidi, D., Turino, C., Potin, B., & Ferretti, G. (2016).Encore ubaye: earthquake swarms, foreshocks, and aftershocks in the southern french alps.Bulletin of the Seismological Society of America,106(5),2244–2257.

Specific comments

Introduction L21-22 : This sentence is referring to a debate about the "seismicity distribution in the Alps". This is vague, please explain in more detail what is the issue (see comments above).

Geological History This section should include geologic and seismotectonic details, in particular a paragraph describing the seismicity distribution and its origin according to previous studies.

Method I would suggest to split this part into "data" and "method".

Technical corrections

L8 : Please, replace "varying seismicity distribution" by "varying spatial seismicity distribution".

L18 : Please explicit the acronym URG.

L24 : Please, indicate the type of model you are referring to.

L26 : Please replace cross-correlation by "link" or "relation" or "correlation"

L196 : The sentence "Viscosities for the lithospheric mantle tend to be between 19 –23 log10Pa s and for the lower crust between 21 –23 log10Pa s with both largely aseismic across the region" is difficult to understand, please rephrase it.

L225 : I would suggest replacing "thereof to" by "on".

L310 : "In thick felsic crustal regions that also lie above a weak lithospheric mantle,

such as the crustal root of the orogen, maximum depths of seismicity are significantly shallower than on the forelands and as such maximum temperatures of seismicity are also significantly lower at ∼350 °C". This sentence is difficult to understand, please rephrase it.

L331 : The end of the sentence ("and") has to be corrected.

Fig1 : Plotting the seismicity on this map would be nice.

Fig 2 and/or 3 : indicate in the caption that numbers correspond to density values.

---

## Referee Comment (RC2) · Anonymous Referee #2 · 9 Mar 2021

The manuscript compares the distribution of earthquakes in a ca. square-shaped region centered on the Alps, and discusses the spatial correlation with a rheological model based on observations. The manuscript is relatively short, ca. 350 lines, with 1 table and 10 figures, no supplement.

While assessing the manuscript, three major concerns crystallized which raise the question on the suitability of this work for publication. These are the following.

1) Area of study. As Figure 1 shows, the area of study does not include the entire Alps. The majority of the Western Alps are left out, and the eastern termination are also left out. I find this is an issue as there are earthquakes in both of these areas and they would add to the rheological discussion as well. A broader, or different shape chosen for the study would justify the title and improve the discussion by a lot. The current map

makes it awkward to really label this work "Alpine".

2) Seismicity. The seismicity data shown in this paper is from the ISC catalogue. This catalogue is known to have drawbacks compared to more detailed local/regional/national catalogues in the Alpine area. This should be discussed in detail, uncertainties are a crucial element of such an analysis. I'm afraid the selection criteria chosen by the authors (lines 131 onwards) removed the majority of events from the map. Moreover, several events are close to the border of the study area – are there edge effects that affect the mechanical analysis? By the way, the majority of the 4405 chosen events are in the Apennines. To illustrate the problem of data selection, a question: can figure 9 (seismic density and maximum depth of events) be interpreted without discussing magnitude of completeness?

3) Only incremental advance. By comparing the proposed figures to those in recent publications of the lead authors, there is a large overlap. Spooner et al. 2019 Solid Earth, as well as Spooner et al. 2020 Global and Planetary Change already include numerous figures of this study. Namely: figure 1, figure 2, figure 3, the data for figure 4, a precursor of figure 5, precursor of figure 6. Figure 9a is a representation of ISC data. Figure 10a is a new way of comparison but similar to the 2020 paper, figure 10b is from another paper. This leaves only Figures 7 and 8 as new. Is this sufficient to publish a paper?

---

## Referee Comment (RC3) · Anonymous Referee #3 · 12 Mar 2021

The authors implement a strength model for the Alpine area and surroundings and compare the results obtained with seismicity distribution. The main novelty of this study is the new input parameter (thermal model) used to calculate the strength and viscosity variations. I think that before the publication, the discussion should be improved providing some clarifications (see detailed suggestions below), referring to the other possible causes of strength variations (besides crustal/lithospheric thickness and temperature), and referring more to previous studies that afforded similar items. The authors interpret variations of the strength and parameters correlated as function of temperature and crustal thickness changes. To this purpose, we should consider that in the model an uniform stain rate is assumed and lateral variation of rheology is not included. These parameters could influence the strength variations and their possible

effects should be discussed. Furthermore, since the Alps and surrounding areas are tectonically active, thermal steady state conditions are likely not present. The authors refer to the possible effects of the slab as well as of the fluids, but processes such as exhumation/erosion/sedimentation can affect the thermal field, especially that of the sedimentary layer. The authors should also specify if they calculated the strength for compressional or extensional stress conditions. Other specific suggestions are below: Section 2 Method: Line 120-125: The authors state referring to the Peierls creep mechanism: "however this was found to not affect the ductile strength of the plate..." What do you mean precisely? They cited Katayama and Karato (2008), but this article refers to an experiment on olivine under water satured conditions, which may not represent the conditions of the study area. However, other experiments (e.g., Demouchi et al., 2013) derived the Peierls creep mechanism on 'dry' olivine as well. There is a recent thermal model of the European lithosphere of Limberger et al., 2018 (Global and Planetary Change) with which the authors can compare their results. In Table 1, the authors display the rheological paramters used, but they do not specify the rocks' conditions (dry or wet), except for the sediments. Equation 3: Please, add a reference to the equation of the effective solid viscosity. Section 3 Results: About the strength results displayed in Fig. 5, it would be better to display the two figures using the same le range of values, possibly with another color scale (the one in use is too dark) to better compare them. At the moment, the lithospheric strength looks almost equal to the crustal strength. Lines 140-155: It would be intresting to correlate the ratio crustal/mantle strength with crustal thickness and temperature to better understand which of the two parameters influences more the strength. Line 165: 'The distribution of seismic event epicentres in the southern foreland strongly correlates spatially with the computed integrated lithospheric strength (Figure 5a) and not with crustal strengths,...' This is hard to say, according to the colour scale used for Figure 5. Furthermore, if the earthquakes occur in the crust, their distribution should correlate more with the crustal strength variations. Line 175: 'all cross sections show that the majority of seismicity occurs within the strongest region of the upper crust ($\sim$ 1 GPa),'

I do not think that you can link the depth of seismicity with a strength value (∼ 1 GPa), since this value is derived from a model based on assumptions, such as a fixed strain rate. Line 183 up to the end of the section: Since, as expected, there is a strong correlation between lateral strength and viscosity variations, I suggest to discuss these results together. Section 4.1 Mechanical strength: The concept that a thick crust (e.g., that one characterizing the orogens) retains more strength than the mantle lithsophere has been also discussed in previous studies (e.g., Tesauro et al., 2009, Tectonophysics for Europe and more recent studies 2 on global and regional scale). About the relationship between crustal thickness, temperature, and integrated strength check also Mareschal and Jaupart, 2013, (Tectonophysics). Lines 215-218: A lower geothermal gradient can result also in an increase of the maximum depth seismicity, due to the deepening of the BDT, and not necessarily in 'less seismicity'. Section 4.2 Relation to sesimicity: The location of seismicity at the boundaries of tectonic features having different rigidity/strength has been already observed in previous studies that the authors can check (e.g., Craig et al., 2011, Geophys. J. Int; Sloan et al., 2011, Geophys. J. Int.; Tesauro et al., 2015, G3). Lines 275-278: The presence/absence of decoupling conditions are more intuitive looking at the profiles of the strength variations than at those of viscosity variations. About the seismicity depth: it can be influenced by the presence of fluids, as in case of the Molasse basin, where the maximum depth is close to that of the Moho (check the study of Deichmann, 1992, Phys. Earth Planet.), besides by the strain rate (a higher strain rate than the one assumed by the authors would increase the BDT depth). Then, the temperature is not the only parameter that influences the seismicity depths.

Please also note the supplement to this comment:
https://se.copernicus.org/preprints/se-2020-202/se-2020-202-RC3-supplement.pdf

---

## Author Comment (AC1) · 19 May 2021

1• One concern is the lack of discussion regarding the representativeness of the seismic catalog used in the study. Although the authors have selected the events according to the location error and the magnitude, the limit of the catalog extracted from ISC and the potential bias linked to the seismic station distribution aren't discussed. This should be addressed, in order to be able to compare the modeled strength of the lithosphere with the epicenters and the depth distribution of the events.

We thank the reviewer for their suggestion. This has been dealt with through the addition of a 'Seismicity Catalogue' section from lines 137 – 167. Two new figures (figures 5 and 6) have also been added to show how the ISC catalogue compares to regional catalogues and the magnitude of completeness.

2• More generally, it's also essential to add a synthesis of the knowledge of the factors controlling the seismicity localization in the area, based on previous studies (e.g. Schmid, S. M., and Kissling, E. (2000), The arc of the western Alps in the light of geophysical data on deep crustal structure, Tectonics, 19( 1), 62– 85, doi:10.1029/1999TC900057; Singer, J., Diehl, T., Husen, S., Kissling, E. and Duretz, T. Alpine lithosphere slab rollback causing lower crustal seismicity in northern foreland. Earth and Planetary Science Letters, 397, pp.42-56, https://doi.org/10.1016/j.epsl.2014.04.002, 2014; Thouvenot, F., Jenatton, L., Scafidi, D., Turino, C., Potin, B., & Ferretti, G. (2016).Encore ubaye: earthquake swarms, foreshocks, and aftershocks in the southern french alps.Bulletin of the Seismological Society of America,106(5),2244–2257) to the ms. Actually, it would help to better contextualize the study and highlight its importance.

We thank the reviewer for this suggestion and have now added a number of paragraphs into the 'Geological and Seismic History' section of the manuscript at lines 67 – 91 in order to rectify this.

3• A large part of the description of the results and the discussion refers to the spatial variability of several parameters (viscosity, integrated strength, maximum depth of seismicity, etc.). These sections are however difficult to follow because this spatial distribution isn't explicit enough with respect to the areas of interest. Using systematically the annotation in the figures (AS, vo, urg,. . ..), or naming clearly the regions would greatly help to understand the reasoning of the authors. E.g. in the sentence L314-316 (" We do however note regions where the maximum temperature of seismicity greatly exceeds 600∘C, corresponding to the presence of both actively subducting and previously subducted slabs, shown as high velocity featuresat a depth of 100km (Figure 10b) from a recent shear wave velocity model of the region (El-Sharkawy et al., 2020)"), it's unclear which regions the authors are referring to.

We thank the reviewer for bringing this to our attention. This has now been changed throughout the text so that all locations mentioned correspond to an annotation used within the figures. As such, additional captions were also added to the cross sections in figures 8 and 10.

4 • L21-22. This sentence is referring to a debate about the "seismicity distribution in the Alps". This is vague, please explain in more detail what is the issue (see comments above).

We thank the reviewer for pointing this out. We feel that the very introduction to the work benefits from the use of such a general statement, however to address this in detail, new paragraphs from line 67 – 91 have been added.

5 • Geological History, This section should include geologic and seismotectonic details, in particular a paragraph describing the seismicity distribution and its origin according to previous studies.

As mentioned in prior responses this change has been made. The section is now renamed 'Geological and Seismic History' and new sentences have been added from lines 62 – 91.

6 • Method I would suggest to split this part into "data" and "method".

We thank the reviewer for this suggestion and have renamed this section '2 Workflow' and split it into subsections '2.1 Data' and 2.2 Method'.

7 • L8 : Please, replace "varying seismicity distribution" by "varying spatial seismicity distribution".

Thanks for pointing this out, this has now been changed.

8 • L18 : Please explicit the acronym URG.

This has been changed, in the abstract as suggested and also in the main body of the manuscript at line 229.

9 • L24 : Please, indicate the type of model you are referring to.

We thank the reviewer for pointing out this lack of clarity. It refers to seismic focal mechanisms and the manuscript has been changed to reflect this.

10 • L26 : Please replace cross-correlation by "link" or "relation" or "correlation".

Thanks for this suggestion, this has now been changed.

11 • L196 : The sentence "Viscosities for the lithospheric mantle tend to be between 19 –23 log10Pa s and for the lower crust between 21 –23 log10Pa s with both largely aseismic across the region" is difficult to understand, please rephrase it.

We thank the reviewer for pointing out the lack of clarity. This sentence has now been split into two sentences and rephrased to add clarity at lines 262 - 264.

12 • L225 : I would suggest replacing "thereof to" by "on".

Thanks for this suggestion, this has now been changed.

13 • L310 : "In thick felsic crustal regions that also lie above a weak lithospheric mantle, such as the crustal root of the orogen, maximum depths of seismicity are significantly shallower than on the forelands and as such maximum temperatures of seismicity are also significantly lower at ~350 ◦C". This sentence is difficult to understand, please rephrase it.

We thank the reviewer for bringing this to our attention. This sentence has now been removed, as upon review we found that it did not contribute meaningfully to the crux of the paragraph.

14 • L331 : The end of the sentence ("and") has to be corrected.

Thanks for pointing this out, this has now been corrected.

15 • Fig1 : Plotting the seismicity on this map would be nice.

We thank the reviewer for this suggestion, this has change has been made.

16 • Fig 2 and/or 3 : indicate in the caption that numbers correspond to density values.

We thank the reviewer for bringing this to our attention. This was already present in the caption of figure 3 but has now also been added to the figure 2 caption.

---

## Author Comment (AC2) · 19 May 2021

1 • Area of study. As Figure 1 shows, the area of study does not include the entire Alps. The majority of the Western Alps are left out, and the eastern termination are also left out. I find this is an issue as there are earthquakes in both of these areas and they would add to the rheological discussion as well. A broader, or different shape chosen for the study would justify the title and improve the discussion by a lot. The current map makes it awkward to really label this work "Alpine".

The study area utilised within this work is constrained by the limits of the structural and thermal models that it draws from (Spooner et al., 2019; Spooner et al., 2020). Presently high resolution datasets, that would allow for the inclusion of the small remaining portions of the very western and eastern Alps that are not included, are unavailable. In order to make this clearer, new sentences have been added at lines 105 – 109.

2 • Seismicity. The seismicity data shown in this paper is from the ISC catalogue. This catalogue is known to have drawbacks compared to more detailed local/regional/national catalogues in the Alpine area. This should be discussed in detail, uncertainties are a crucial element of such an analysis. I'm afraid the selection criteria chosen by the authors (lines 131 onwards) removed the majority of events from the map. Moreover, several events are close to the border of the study area – are there edge effects that affect the mechanical analysis? By the way, the majority of the 4405 chosen events are in the Apennines. To illustrate the problem of data selection, a question: can figure 9 (seismic density and maximum depth of events) be interpreted without discussing magnitude of completeness?

We understand the reviewer's concerns about the ISC catalogue. In response to these and the suggestions of other reviewers we have added a thorough analysis of the catalogue with comparison to local catalogues (figure 5), as well as an analysis of the catalogue completeness (figure 6) along with a new 'Seismicity Catalogue' section in the text from lines 137 – 167.

3 • Only incremental advance. By comparing the proposed figures to those in recent publications of the lead authors, there is a large overlap. Spooner et al. 2019 Solid Earth, as well as Spooner et al. 2020 Global and Planetary Change already include numerous figures of this study. Namely: figure 1, figure 2, figure 3, the data for figure 4, a precursor of figure 5, precursor of figure 6. Figure 9a is a representation of ISC data. Figure 10a is a new way of comparison but similar to the 2020 paper, figure 10b is from another paper. This leaves only Figures 7 and 8 as new. Is this sufficient to publish a paper?

This work does indeed build off of previous works that have constrained the structure and temperatures of the lithosphere in the region, such that lithospheric strength can now be calculated and discussed within this work. Should the figures of the study area, the crustal thicknesses and thermal field therefore remain excluded from this work, despite their clear relevance to the discussion of the manuscript? We and perhaps the majority of the geoscience community would say that is not the case. We would also argue that a figure that is 'a new way of comparison but similar' demonstrates a clear understanding of the scientific process and how findings improve and evolve as studies progress in a field. We also struggle to see what the reviewer could be referring to from our previous work as a 'precursor' of figure 5 (now figure 7) and figure 6 (now figure 8) given that these represent the entirely new strength results generated in this work. Most of the points made here by the reviewer seem to abjectly dismiss figures in this work, without due consideration (or understanding) of the value that they bring to the discussion. Nevertheless, as a result of the numerous constructive comments offered by two other reviewers, the manuscript has been significantly expanded with the addition of multiple new figures and sections in the text, to broaden the discussion and improve the scope of the manuscript.

---

## Author Comment (AC3) · 19 May 2021

1 • The authors interpret variations of the strength and parameters correlated as function of temperature and crustal thickness changes. To this purpose, we should consider that in the model an uniform stain rate is assumed and lateral variation of rheology is not included. These parameters could influence the strength variations and their possible effects should be discussed. Furthermore, since the Alps and surrounding areas are tectonically active, thermal steady state conditions are likely not present. The authors refer to the possible effects of the slab as well as of the fluids, but processes such as exhumation/erosion/sedimentation can affect the thermal field, especially that of the sedimentary layer.
We thank the reviewer for these helpful suggestions. We have now fully addressed all of these points in the newly added 'Workflow Limitations' section from lines 403 – 425.

2 • The authors should also specify if they calculated the strength for compressional or extensional stress conditions.
We thank the reviewer for pointing out this omission, it has been calculated in a compressional regime and this has been added to the manuscript at line 190.

3 • Section 2 Method: Line 120-125: The authors state referring to the Peierls creep mechanism: "however this was found to not affect the ductile strength of the plate…" What do you mean precisely? They cited Katayama and Karato (2008), but this article refers to an experiment on olivine under water satured conditions, which may not represent the conditions of the study area. However, other experiments (e.g., Demouchi et al., 2013) derived the Peierls creep mechanism on 'dry' olivine as well.
We thank the reviewer for their suggestion. He/she is correct in stating that we refer to a specific form of low temperature plasticity as derived from experiments on wet olivine. Our choice for a wet rheology stems from the following reason. Peierls creep results in a weakening of the plastic strength of the mantle rock at higher stress, which could potentially influence the depths at which the transition between frictional brittle behaviour and ductile deformation occurs. As such, the BDT would be located at shallower depths than those computed based on conventional power law creep. Recent studies (e.g. Katayama, 2021) have highlighted the sensitivity of lithospheric strength and modes of deformation to the effect of water. This is particularly relevant while addressing the sensitivity Peierls creep shows to wet conditions. The main implication here is that the effective role of Peierls creep to weaken the strength of the mantle would be higher in the presence of fluid, therefore by considering a bulk wet rheology for the most abundant mantle mineral, e.g. olivine. In this regard, a main limitation is the "upscaling" of currently adopted flow laws from the size of the sample in the laboratory to the scale of the lithosphere, where also we lack any control on the effective boundary conditions (the latter are controlled in the laboratory). There is also no evidence of scale invariant behaviour of power law creep rheologies (whether diffusion, dislocation or Peierls), given their exponential dependence on the applied strain rate. The latter parameter also differs within at least ten orders of magnitude between laboratory and natural conditions. Therefore, such models can only propose end-member study cases. Back to the "wet versus dry" bulk rheology issue, within the range of uncertainties in flow law parameterization, we attempted in our manuscript to quantify how robust the modelling results would have been to the additional weakening from low temperature plasticity. Therefore, we tested an end member model where these effects are "maximized", thus our choice for a wet rheology, and we found that this additional deformation mechanism did not affect the main implications derived from our study. This would also be valid if considering a dry rheology, for which the weakening would be even less pronounced.

When stating that, "... this was found to not affect the ductile strength of the plate…", we are showing that Peierls creep did not affect the spatial distribution of the brittle to ductile transition, where the strength contrast is controlled by Peierls creep mechanism at those levels.

4 • There is a recent thermal model of the European lithosphere of Limberger et al., 2018 (Global and Planetary Change) with which the authors can compare their results.
We thank the reviewer for this suggestion. As the thermal field was not generated within this work (see Spooner et al., 2020) we did not think it relevant to discuss differences between thermal field at length in the discussion. However, we have added a paragraph in the 'Structural Model and Thermal Field' section from lines 124 - 136 to explain why the thermal field we have used is preferential to Limburger et al. (2018) and compare the differences between the two.

5 • In Table 1, the authors display the rheological paramters used, but they do not specify the rocks' conditions (dry or wet), except for the sediments.
We thank the reviewer for pointing out this omission. Table 1 has now been updated to reflect that the other parameters were dry.

6 • Equation 3: Please, add a reference to the equation of the effective solid viscosity.
We have some problems in following the reviewers reasoning here. The solid viscosity which he/she is referring to is not a material property, rather an effective parameter that is derived to recast the main constitutive law for secondary creep linking (differential) stress to (differential) strain rate in a more concise manner, thereby integrating all non-linear dependencies (described in equation 2 in the main text). This said, we are unsure which reference, if any, could apply here.

7 • Section 3 Results: About the strength results displayed in Fig. 5, it would be better to display the two figures using the same le range of values, possibly with another color scale (the one in use is too dark) to better compare them. At the moment, the lithospheric strength looks almost equal to the crustal strength.
As the range of lithospheric strengths ($0.9 \log_{10}$ Pa m) is almost 4 times larger than the range of crustal strengths ($0.25 \log_{10}$ Pa m), plotting them on the same scale whilst showing the heterogeneity of crustal strengths is not possible. The intent of these figures (now figure 7) is to show lithospheric and crustal strength heterogeneity and each has been scaled to highlight this. We have however changed the colour scale slightly from before in order for it to appear less dark, as per the reviewer's suggestion, which we are thankful for. For a common scaling showing how much the crust contributes to lithospheric strength please see, Figure 8, which we have used in the manuscript to discuss how the crust provides very little of the lithospheric strength except in the orogenic root, with the upper lithospheric mantle providing the majority of lithospheric strength in both forelands. We have also added a new figure (figure 11) which shows the crustal contribution to overall strength making these comparisons even easier.

8 • Lines 140-155: It would be intresting to correlate the ratio crustal/mantle strength with crustal thickness and temperature to better understand which of the two parameters influences more the strength.

As mentioned in the above response, figure 11 has been added in order to shows the crustal contribution to the overall lithospheric strength. This has also been discussed in a significantly reworked paragraph in the 'Mechanical Strength' section at lines 278 -289.

9 • Line 165: 'The distribution of seismic event epicentres in the southern foreland strongly correlates spatially with the computed integrated lithospheric strength (Figure 5a) and not with crustal strengths,…' This is hard to say, according to the colour scale used for Figure 5. Furthermore, if the earthquakes occur in the crust, their distribution should correlate more with the crustal strength variations.

As has been mentioned in a prior response, the colour scaling was scaled to highlight the relative strengths of either the crust or lithosphere, as what is important is whether the crust is strong compared to other places in the crust. To the reviewers second comment, what we are trying to demonstrate is that that is not always the case, with plate dynamics having an impact, resulting in lots of seismicity in certain regions where the crust appears homogenously strong, as the integrated lithospheric column is weak there due to representing the edge of a rigid and rotating indenter.

10 • Line 175: 'all cross sections show that the majority of seismicity occurs within the strongest region of the upper crust (~ 1 GPa),' I do not think that you can link the depth of seismicity with a strength value (~ 1 GPa), since this value is derived from a model based on assumptions, such as a fixed strain rate.

We thank the reviewer for bringing this to our attention. We have addressed this by making it clear in the 'Results' section that these strength values are merely values derived from the calculations undertaken in this workflow (see lines 242, 245 and 247). In addition, we have added a further section on 'Workflow Limitations' (see lines 403 – 425) to discuss the impact that depth dependent strain rates may have and that not enough data is available at present in order to utilise them across the whole study area.

11 • Line 183 up to the end of the section: Since, as expected, there is a strong correlation between lateral strength and viscosity variations, I suggest to discuss these results together.

We agree with the reviewer that there is a clear link between the lithospheric strength and the mantle viscosity which has already been emphasised in the text when discussing the mantle viscosities. However, there is also a very strong link between the lateral strength variations and the strength cross sections. As such we have opted to retain the original layout as we feel it already represents the most efficient way to talk through the results of the study.

12 • Section 4.1 Mechanical strength: The concept that a thick crust (e.g., that one characterizing the orogens) retains more strength than the mantle lithsophere has been also discussed in previous studies (e.g., Tesauro et al., 2009, Tectonophysics for Europe and more recent studies 2 on global and regional scale). About the relationship between crustal thickness, temperature, and integrated strength check also Mareschal and Jaupart, 2013, (Tectonophysics).

We thank the reviewer for this suggestion. We have now included these references in a significantly reworked paragraph in the 'Mechanical Strength' section at lines 278 – 289.

13 • Lines 215-218: A lower geothermal gradient can result also in an increase of the maximum depth seismicity, due to the deepening of the BDT, and not necessarily in 'less seismicity'.

Whilst we agree with the reviewer that this is of course a possibility, it is not observed within our results as almost no seismicity occurs beneath the upper crust in the orogen. Instead, the

lower geothermal gradient results in a stronger crust which appears to result in less seismicity. We have reworded this section and moved it to the 'Relation to Seismicity' section at lines 298 - 303 in order to add clarity.

14 • Section 4.2 Relation to sesimicity: The location of seismicity at the boundaries of tectonic features having different rigidity/strength has been already observed in previous studies that the authors can check (e.g., Craig et al., 2011, Geophys. J. Int; Sloan et al., 2011, Geophys. J. Int.; Tesauro et al., 2015, G3).

We thank the reviewer for bringing these studies from other regions to our attention and have incorporated them into the manuscript at line 312.

15 • Lines 275-278: The presence/absence of decoupling conditions are more intuitive looking at the profiles of the strength variations than at those of viscosity variations.

We thank the reviewer for bringing this to our attention. We agree and have made some changes to the wording in the paragraph at lines 345 – 356 in order to reflect this.

16 • About the seismicity depth: it can be influenced by the presence of fluids, as in case of the Molasse basin, where the maximum depth is close to that of the Moho (check the study of Deichmann, 1992, Phys. Earth Planet.), besides by the strain rate (a higher strain rate than the one assumed by the authors would increase the BDT depth). Then, the temperature is not the only parameter that influences the seismicity depths.

We thank the reviewer for bringing this work to our attention. We have incorporated it into the discussion at lines 400 – 402. Unfortunately, we cannot comment on the likelihood of fluid flow as it has not been modelled within this work. We do however point out that the location of this deeper observed lower crustal seismicity that occurs at higher than expected temperatures aligns well with the location of alpine slabs, suggesting that the slabs might provide a regional cooling effect due to not being in thermal equilibrium that we have not accounted for. Other papers cited in our work (Singer, J., Diehl, T., Husen, S., Kissling, E. and Duretz, T. Alpine lithosphere slab rollback causing lower crustal seismicity in northern foreland. Earth and Planetary Science Letters, 397, pp.42-56, https://doi.org/10.1016/j.epsl.2014.04.002, 2014) also pose slab rollback as a different hypothesis for this deep seismicity.